# Social Exclusion, Surveillance Use, and Facebook Addiction: The Moderating Role of Narcissistic Grandiosity

**DOI:** 10.3390/ijerph16203813

**Published:** 2019-10-10

**Authors:** Myungsuh Lim

**Affiliations:** Department of Business Administration, Sangji University, Wonju 26339, Korea; mslim@sangji.ac.kr

**Keywords:** social exclusion, surveillance, Facebook addiction

## Abstract

One hundred and eighty-eight participants completed the online questionnaire with items on demographics (age and gender), social exclusion, surveillance use, Facebook addiction, and narcissistic grandiosity. The findings showed that social exclusion was positively associated with Facebook addiction (B = 0.237, *p* < 0.001) and surveillance use was significantly positively associated with Facebook addiction (B = 0.211, *p* < 0.01). The surveillance use of Facebook was found to be a significant mediator between the risk of social exclusion on Facebook and Facebook addiction (B = 0.054, CI [0.20, 0.113]). Narcissistic grandiosity significantly moderated the associations between social exclusion and Facebook addiction (B = 0.079, *p* = 0.012). These findings suggest that the risk of social exclusion could serve as facilitator of Facebook addiction depending on narcissistic grandiosity.

## 1. Introduction

Although the use of Facebook has helped increase social capital and build relationships [1], “unfriending” on Facebook leads to negative emotions [2], promotes social media ostracism, and inhibits well-being [3]. A root cause of these negative effects is that Facebook is not always used to maintain or develop relationships. For example, voyeurism, a positive driver of Facebook use, is a form of social surveillance that fulfills the need to belong but attempts to avoid negative feelings and experiences; those who engage in it do not aim to maintain relationships with those they observe [1]. Moreover, Facebook is a platform that makes it easy to self-promote, allowing others to easily surveil the extensive content produced.

On the other hand, cyber-ostracism, which causes social exclusion on the Internet, has been found to be a threat to fundamental human needs, such as the needs of belonging and high self-esteem [4]. The negative effects of feeling excluded are not significantly different whether they arise in the context of a remote communication method or through in-person situations [5]. To cope with their online social exclusion, Facebook users may search for social information to restore their affiliation with other users [6], which facilitates their addiction to Facebook. On the other hand, people with narcissistic tendencies may be more likely to cope with social exclusion in order to maintain their ego-centric aspect [7], although the affiliation need is low [8].

Previous studies have shown that Facebook overuse to fill the need to belong can lead to Facebook addiction [9]. However, there is a lack of research identifying the mediation variables in the process of social exclusion and Facebook addiction and investigating social network attachment in social exclusion situations that do not meet the need to belong [10]. In addition, the self-presentation need [11] and the need for admiration [9] have been cited as the main reasons for using Facebook. Narcissism, in particular, has drawn attention as a major personality characteristic associated with problematic Facebook use [12]. The present research, therefore, investigated the following hypothetical relationships: (1) whether social exclusion on Facebook is positively related to addiction to Facebook; (2) whether those with a narcissistic personality trait have a greater tendency to become addicted to Facebook as a way to avoid social exclusion; and (3) whether the use of Facebook for surveillance in response to social exclusion leads to Facebook addiction.

## 2. Hypotheses Development

### 2.1. Social Exclusion and Facebook Addiction

One of the main reasons that people use the Internet is to reveal themselves and to gain a sense of belonging [13]. A human being is innately motivated to belonging as a fundamental impetus to form and maintain relationships with others [14]. Social exclusion, which undermines the social affiliation that occurs on Facebook, happens when people are restricted from producing content or are unable to receive responses from others [15]. Addiction to social network services (SNS) is classified as Internet-related relationship addiction [16]. While whether SNS addiction should actually be classified as addiction is still debated, it does display relevant symptoms similar to other substance and behavioral addictions, such as mood-repair experience and salience (e.g., full attention to usage) [17]. Facebook is the leading SNS and has been shown to have strong correlations with symptoms classified as Internet addiction [18]. Addiction to Facebook interferes with vital activities in life, and virtual relationships dominate actual relationships [19].

Social exclusion may lead to Facebook addiction for the following reasons. First, social exclusion causes social anxiety. Socially anxious people are prone to becoming immersed in the Internet. According to the social skill model, high levels of social anxiety lead to the increased use of the Internet in order to elevate the sense of self-worth [13]. The fear of social exclusion increases the user’s social risk, which leads to the pathological use of the Internet [13].

Second, users who feel socially excluded are motivated to restore their social affiliation, which may lead to Facebook addiction. They want to feel that they belong to a group, which includes affiliation and companionship [20]. Considering that the aim to identify one’s social identity within the group to which one belongs and fulfill the need to belong is a significant predictor of the increase in SNS usage [12], social exclusion may make users more obsessed with SNSs. Hence, the lack of relatedness is connected to Facebook addiction [21].

Third, a primary driver of social media use is the fear of being excluded and forgotten, which increases the frequency of Facebook use [10,22]. For example, those snubbed by others on the phone attach to social networks to regain the feeling of social inclusion [10]. Both positive and negative feedback from other users affect Facebook users’ well-being, and so does failure to receive feedback because FOMO (Fear of Missing Out) increases the Facebook-related stress stemming from not being popular or involved with peers on Facebook [23].

**Hypothesis 1** **(H1).***The higher the degree of social exclusion on Facebook is, the greater the Facebook addiction is*.

### 2.2. Social Exclusion and Surveillance Use

Interpersonal electronic surveillance refers to the use of communication technologies to observe another people’s behavior both online and offline. The subjects of surveillance can be close friends, romantic partners, business associates, and family members [24]. Surveillance also allows users to maintain a healthy interpersonal relationship by constantly engaging with others [25]. However, it also compromises privacy settings [24]. Because SNSs, including Facebook, are repositories of messages, pictures, and videos, they provide an environment that facilitates surveillance. In the present study, surveillance concerns the use of surveillance in horizontal relationships between ordinary people [24].

Social exclusion may promote surveillance for the following reasons. Social exclusion promotes the heightened attention to social information and the processing of social information [26]. Social exclusion motivates both the need for belonging and the use of surveillance to cope with the sense of isolation [27] for successful social survival. Social exclusion induces the feeling of not belonging to a group, which generates negative effects, such as anxiety, loneliness, and antisocial behaviors [14]. Users who fear social exclusion continuously monitor their social status to maintain their feeling of social inclusion [28]. This behavior reduces the sense of loneliness caused by social exclusion [29], thus resolving social anxiety. This social monitoring system extends to the social environment and social opportunities [27]. This goal-directed behavior is motivated by the need for social affiliation. The sense of belonging is the innate, evolutionarily adaptive human motive [14]. People seek social information to classify the relationship with another party, to distinguish information that should be strategically stored for the relationship, and to verify the extent to which the social relationship is related to the event, when forming attribution [6]. In addition, the level of information processing at this time can be influenced by the degree of concern about the relationship [6]. Similarly, the social exclusion that users identify on Facebook forces them to face their uncertain social status [30] and to pursue goal-relevant behavior to restore their sense of social affiliation. Users who feel socially excluded are sensitive to social information and actively process it [6]. Based on the above discussion, the following hypothesis is stated:

**Hypothesis 2** **(H2).***The higher the degree of social exclusion is on Facebook is, the higher the use of Facebook for surveillance is*.

### 2.3. Surveillance Use and Facebook Addiction

Surveillance use may lead to Facebook addiction for the following reasons. First, remote technologies such as the Internet increase the effect of disinhibition effect on users [31]. Therefore, the effects of social exclusion online to users can be just as powerful as those of social exclusion offline [32]. Because social exclusion online motivates the need to belong [33], Facebook is frequently used for surveillance [34]. People who feel lonely because of social exclusion exhibit higher levels of social monitoring and scanning the environment for social cues [27].

Second, the function of mutual surveillance on Facebook enhances addiction. Facebook’s surveillance function keeps people informed about their connections and personal information [35]. Therefore, when social exclusion occurs, people can access social information on Facebook, which increases the number of alternative actions that can compensate for the lack of social relationships. Users pursue surveillance gratification by viewing photographs or newsfeeds on Facebook, which may be associated with the fear of “missing out” [36]. Individuals who have experienced social exclusion may become addicted to Facebook by using Facebook’s informative functions to overcome the sense of isolation. Based on the above discussion, the following hypothesis is stated.

**Hypothesis 3** **(H3).***The higher the use of Facebook for surveillance is, the greater Facebook addiction is*.

### 2.4. The Mediating Role of Surveillance Use

This study supports recent findings that anxiety about social exclusion (e.g., FOMO) causes surveillance use [37]. As discussed, the FOMO and the need to belong motivate people to overcome uncertainty about their social positions by seeking social information, which encourages surveillance use of Facebook. A lack of a sense of social belonging thus can result from being addicted to Facebook through passive, continuous monitoring of relationships rather than being stimulated to actively form relationships. The Facebook environment also promotes disinhibition in individuals, and the mutual surveillance function of Facebook leads to addiction among those using it for surveillance. Based on this theoretical review, the following hypothesis is proposed:

**Hypothesis 4** **(H4).***Surveillance use mediates the relationship between social exclusion and Facebook addiction*.

### 2.5. Moderating Effects of Grandiosity on Social Exclusion and Facebook Addiction

There may be individual differences in the responses to social exclusion [38]. Narcissistic grandiosity is characterized by arrogant, conceited, and domineering attitudes and behaviors based on maladaptive self-enhancement. Unlike others, narcissists have heightened needs for recognition and admiration [39]. In addition, individuals who exhibit the characteristics of narcissistic grandiosity are highly concerned about impression management [40], which frequently occurs on Facebook [41]. If social exclusion occurs on Facebook, the grandiose narcissist will try to find ways to maintain self-enhancement.

Therefore, narcissistic grandiosity may become a variable that has a moderating effect on personal differences in the relation between social exclusion and Facebook addiction. The reasons are as follows. First, the primary reason that narcissists use Facebook is self-exhibition. Therefore, they continuously pursue their intention to use Facebook even in the context of social exclusion. Narcissists have low affiliation needs [8]. Their reason for social networking is not to build social relationships but to present themselves as superior to others [1]. Despite their social exclusion, they can still be addicted to social networking without the surveillance intention and the need to acquire social information about others. Furthermore, they tend to have a high propensity for addiction to SNSs per se [12]. Exhibitionism is a powerful motivator for narcissists to use SNSs [42]. They seek a wide audience rather than social interaction [43]. Their behavior to attract attention becomes stronger when they are socially excluded, which leads to SNS addiction.

Second, a high degree of narcissistic grandiosity promotes unstable self-esteem [44]. Those with unstable self-esteem respond more sensitively to social exclusion than others do [45]. They think that they can stabilize their self-esteem by immersing themselves in online activity [46]. Persons with a high narcissistic tendency and high inconsistency between explicit and implicit self-esteem continually monitor their position because they are insecure and prone to self-doubt [47]. In this case, perceived social anxiety is a major factor in the problematic use of the Internet [13]. In addition, social anxiety is related to low self-esteem [17].

Third, narcissists tend to respond more strongly to social exclusion than others do [48]. They want to avoid the negative result of social exclusion. Narcissists perceive exclusion, such as social rejection, as a serious threat to their ego [48]. Furthermore, users with narcissistic grandiosity are impulsive, which exacerbates their addiction to SNS [49]. Based on the foregoing theoretical review, the following hypothesis is stated.

**Hypothesis 5** **(H5).***The direct effect of social exclusion on Facebook addiction is contingent on narcissistic grandiosity such that the effect is stronger in those who are high in narcissistic grandiosity*.

### 2.6. The Research Model

The current study aims to examine the association between social exclusion and Facebook addiction. The research model derived from the findings of the literature review is based on the following assumptions: (1) a positive association exists between social exclusion as predictor variable (X) and Facebook addiction as the dependent variable (Y); (2) this association is mediated by positive surveillance use as mediator variable (M); and (3) grandiosity as moderator variable (W) moderates the direct associations hypothesized, and associations are stronger in individuals with higher grandiosity.

## 3. Materials and Methods

### 3.1. Participants and Procedure

Upon their agreement to participate in the study, the participants were informed that they would be asked to discuss their personal experiences regarding Facebook. No conflicts of interest were reported in the current study. The participants recruited from CrowdFlower (Figure eight, San Francisco, USA), a crowdsourcing data collection platform, completed the survey (Mean age = 32.64, Standard Deviation (SD) = 23.16; females = 60.6%; n = 188). The area of data collection was limited to inside the United States. The average survey completion time was 3.51 min (SD = 3.07). The online introduction explained the policy on privacy protection to all participants before the survey began. The survey introduction described the monetary reward, and after completing the survey, the participants were paid through virtual accounts. All procedures followed were in accordance with the ethical standards of the responsible institutional committee on human experimentation.

### 3.2. Measures

A 26-item, English-language survey was designed using Qualtrics.com (https://www.qualtrics.com). Before they answered the questions, the survey asked the participants to recall a real interaction with a Facebook friend, such as a romantic partner, co-worker, colleague, virtual friend, etc., whom we named ‘X’. While the participants were recalling the relationship with ‘X’, they indicated the extent of their social exclusion from X, the extent of their use of Facebook for surveillance and the extent of their Facebook addiction. They then checked whether they had the propensity for narcissistic grandiosity (see the Appendix A). Finally, the participants marked gender and age according to their demographic.

Unless otherwise indicated, the measures are based on a 7-point Likert-type scale (1 = strongly disagree, 7 = strongly agree). Social exclusion was measured using four of the six items on the Perceived Risk of Exclusion Scale [50]. Because the two excluded items were assumed to concern social exclusions offline, they were not suitable for the study of social exclusion on Facebook. An example is “I wonder if X might try to avoid me”. The participants responded on scales regarding the extent to which the Facebook user perceived the possible risk of exclusion and avoidance by the Facebook friend. The items were summed (Cronbach’s α = 0.921). Facebook surveillance was measured using four items [51]. These items were selected to assess the extent to which the participants agreed with the behaviors of paying attention to and monitoring the Facebook friend. An example of an item is “I view this person’s profile to monitor his/her interactions and watch out for his/her best interests”. The participant’s responses were averaged (Cronbach’s α = 0.725). Facebook addiction was measured with six items [19]. These items assess addiction in SNS use. An example is “Since I have been on Facebook my grades/success on work have deteriorated/my performance at work is worse”. The participants’ responses were averaged (Cronbach’s α = 0.867). Narcissistic grandiosity was assessed by ten items. These items were used to measure the degree of the propensity for grandiose exhibition according to the Narcissistic Personality Inventory [52], the standard measure of subclinical narcissistic traits. An example is “I really like to be center of attention”. The participants responded on a scale ranging from 1 = “Not at all like me” to 7 = “Very much like me”. Higher scores indicated higher narcissistic grandiosity. The summed items showed good reliability (α = 0.921). Additionally, factor analysis was performed by applying varimax rotation to confirm the validity of the measure. In the results, the set of four factors accounted for 65.348% of the variance. The factor loading of the items ranged from 0.505 to 0.811, and all the items used had acceptable validity.

### 3.3. Statistical Analyses

The hypotheses were tested using model 5 in PROCESS macro [53], which allows statistical direct, indirect, and conditional direct effects to be assessed simultaneously. In testing the indirect effect of X on Y through M, the conditional direct effect of W was accounted for; furthermore, the conditional direct effect of W on the relationship between X and Y was tested while controlling for M. To test the indirect effects, 5000 bootstrapped resamples were used which generated a bias correction of 95% and adjusted confidence intervals (CI). The CIs that excluded zero demonstrated a statistically significant indirect effect. The two indirect effects were tested simultaneously using the same model that was used to test the conditional direct effect. To test the conditional direct effects, the predictor and moderator variables were mean-centered, and the significant conditional direct effects were decomposed by plotting the slopes at ±1 standard deviation [54].

## 4. Results

### 4.1. Preliminary Analyses

First, all the main study variables were tested for non-normality. All measures of skewness (−0.242 to 1.213) and kurtosis (−1.827 to 0.942) were below the cutoff of satisfaction. Second, Mahalanobis distance was used to check for multivariate outliers in the main study variables. Because no participant was found to be a multivariate outlier, the final study sample was 188 participants. Lastly, to decide what, if any, demographic control variables to include in the model testing, the bivariate correlations between the study variables and the demographic characteristics (for age and gender) were examined. The correlations were based on the demographic variable and the study variables. Therefore, age and gender were included as a control variable in all subsequent analyses. The means, standard deviations, and bivariate correlations of the study variables used in the analysis are shown in Table 1. All study variables were intercorrelated in the predicted directions.

### 4.2. Testing the Research Model

The results showing the unstandardized regression coefficients are presented in Figure 1. This study first hypothesized that social exclusion is positively associated with Facebook addiction (H1). Figure 1 shows that social exclusion and Facebook addiction were indeed significantly and positively related, indicating that as social exclusion increased, so did Facebook addiction (B = 0.237, t = 4.508, *p* < 0.001, CI [0.133, 0.341]. Thus, H1 was supported. Next, social exclusion was positively associated with surveillance use (B = 0.259, t = 4.524, *p* < 0.01, 95% CI = [0.146, 0.371]). Thus, H2 was supported. In addition, surveillance use was positively associated with Facebook addiction (B = 0.211, t = 3.160, *p* < 0.01, 95% CI = [0.079, 0.342]). Thus, H3 was supported. This study predicted that the mediation model of social exclusion would be related to Facebook addictive tendencies through Facebook surveillance use (H4). The results showed that the indirect effect of social exclusion on Facebook addiction through Facebook surveillance use was significant (B = 0.054, CI [0.020, 0.113]). That is, when Facebook users reported higher levels of social exclusion, they also reported higher levels of Facebook surveillance use. Similarly, as Facebook surveillance use increased, Facebook addiction also increased. Thus, H4 was supported.

Lastly, this study hypothesized the conditional direct effect of narcissistic grandiosity on the relationship between social exclusion and Facebook addiction (H5). The results revealed that narcissistic grandiosity significantly moderated the relationship between social exclusion and Facebook addiction (B = 0.079, t = 2.526, *p* = 0.012, CI [0.017, 0.140]). Figure 2 illustrates this interaction, showing that there were differences between Facebook addiction at high levels of narcissistic grandiosity, at medium levels of narcissistic grandiosity, and at low levels of narcissistic grandiosity, Facebook users reported the highest levels of Facebook addiction when they also reported high levels of social exclusion. Thus, H5 was supported (see Table 2).

## 5. Discussion

The theoretical contributions of this research are as follows. This study elaborated an earlier study of Filipkowski and Smyth [5] that confirmed that the effects of online social exclusion on anxiety and emotions were as harmful as social exclusion in personal communication. This result of the present study indicated that the recent phenomenon of social exclusion on SNSs, which is highly common, can lead to surveillance behaviors and result in addiction.

The mediational role of surveillance use can be explained by the social connection theory and the environmental characteristics provided by Facebook. According to the social connection theory [55], social exclusion means that the need of social connection is not met. The loss of an SNS relationship increases the stress regarding social exclusion and further immerses the user in SNS use [56]. From an evolutionary perspective, humans have a basic need to attend to resources that provide social connections and satisfy the affiliation motivation [14,57]. On Facebook, user-generated content is stored through various forms, such as location data, status and mood messages, comments to profiles, videos, and pictures [58]. These resources facilitate surveillance and may also be linked to Facebook addiction behaviors.

The present research refers to a negative aspect of SNS use, but other studies assert that the use of social resources through SNSs can improve user well-being [59,60]. Authentic behavior showing true self-presentation on an SNS can act as a predictor of well-being to enhance the longitudinal health of the user. However, it may not offer appropriate social capital, such as social support, to a person in a low state of well-being due to positivity bias in SNS communication [61]. The narcissistic tendencies this study examines as individual differences are characteristics found in SNS personalities [62], and also narcissistic tendencies are those of a disagreeable extrovert [63]. Therefore, narcissists who do not communicate through their true selves may attempt to pursue relational interests because they do not receive sufficient social support in SNSs where positive bias occurs. In this situation, the previous research on cyber-ostracism related to narcissism focused on aggression as a maladaptive behavior caused by social exclusion [4]. These results of the present study demonstrated that the higher the level of narcissistic grandiosity is, the more severe the Facebook addiction is as another form of maladaptive behavior.

The present study has several limitations that may serve as directions for future research. First, this research did not distinguish among the types of social exclusion and relationships (e.g., romantic partners, co-workers, colleagues, and virtual friends) on Facebook. The typical Facebook friends recalled in the survey comprised heterogeneous groups that differed in the strength of their ties, intimacy, and emotional bonding [64]. Distinguishing these friend groups can produce different results. For example, information processing differs according to whether the social exclusion is caused by collective rejection or dyadic rejection [6]. This distinction could also influence Facebook addiction, so it would be meaningful to distinguish the types of rejection that cause social exclusion. In addition, the intensity of the relationships between romantic partners, coworkers and virtual friends differed. For example, according to the relationship type, social exclusion may result in different anti-social behaviors. Offensive behavior due to social exclusion is more frequent in easily replaced relationships than close relationships [65]. For example, in the case of strong ties (e.g., romantic partners) facing increased relational uncertainty, surveillance can be pursued as negative maintenance of romantic partnerships [66]. Meanwhile, the weak ties among some acquaintances (e.g., work colleagues) may benefit more from direct interactions. Thus, under weak ties, Facebook surveillance in social exclusion can be intensified [64]. Therefore, in the present study, the intensity of the relationship could have affected the causal relationship between the research variables. However, this study did not consider social exclusion and the correlations arising from different type of relationships because it evaluated various relationship types simultaneously in the survey condition. Therefore, in future research, the differences in the indirect effects of surveillance use should be distinguished in the relation with social exclusion and Facebook addiction depending on the type of relationship.

Second, demographic variables including individual differences were not used to confirm the correlations in the model, but future research should introduce them as crucial variables. Individuals have different reactions to social exclusion. For example, individuals with feelings of loneliness or a high need for social belonging are likely to monitor social information [27]. Those in closed romantic relationships might seek information about ex-partners through Facebook surveillance to reduce relational uncertainty, which causes emotional distress [67]. Thus, to deeply understand the specific personal relationship situations (e.g., romantic break-ups) that affect Facebook surveillance use and addiction, the current research model should reflect the characteristics of the users’ networks. For example, do other users in Facebook users’ networks actively use SNS in terms of the generation? Are there gender differences in responses to relational problems? The results suggested that females respond more negatively than males to interpersonal stressors such as social exclusion [68]. How important are the formation and dissolution of relationships in the life cycle as marital status varies by individual choice? After all, users’ personal characteristics can affect their Facebook networks. To increase the validity of this research model, therefore, it is necessary to follow changes in the relationships between the variables by including individual difference variables.

Third, attention should be paid to the interpretation of the narcissistic personal trait, which serves as a moderating variable in this study. This study used narcissistic grandiosity to measure the subclinical narcissistic trait rather than the index to evaluate narcissistic personality disorder. These items used by Gentile et al. [69] reflect the tendency of college students to show more narcissistic personal traits than morbidity than did those of previous decade and also limit the ability of the results to track Facebook usage patterns. Even considering narcissistic tendencies as a minor trait, the amount of time a user takes editing on Facebook before intentionally showing themselves [69] could affect the use of Facebook. In the DSM-5 [70], histrionic personality and narcissistic personality overlap in their pursuit of attention and focus on impression management. In this study, the narcissistic tendencies are framed by emphasis on self-enhancement based on grandiose gestures. On the other hand, research on histrionic disorder tendencies on SNSs explores behavior such as exaggeration of the attractiveness of the selfie [71]. However, there is also a study that defines a focus on selfies as narcissistic behavior [72]. Therefore, it is necessary for future SNS-user research to more clearly discuss differences between the behaviors exhibited by the users with these disorders.

Fourth, to generalize the research results, it is necessary to attempt to reduce future sampling errors. First of all, the number of samples should be increased. Fewer samples were used compared with the items used. Attention, therefore, should be paid to the interpretation of the results. As the sample size decreases, the standard deviation may increase, resulting in less accurate results and a non-response bias as some subjects might not be able to complete the survey. Additionally, if, in a small sample, only those interested in a topic participate in a survey, voluntary response bias occurs, increasing the chances of skewness in the responses [73]. Future studies, therefore, should have more survey participants to be able to generalize the measurement model. Also, this study validated the model limited to Facebook. To generalize the research results, it is necessary to confirm them by collecting variables from various SNS platforms.

Fifth, the constructs of the study were dependent on self-reported data, which are prone to bias [74]. Hence, future researchers are encouraged to employ objective measures to collect data from various sources to strengthen the reliability of the findings.

## 6. Conclusions

The purpose of this study was to identify the mediation role of surveillance use and the moderating effect of narcissistic grandiosity in the relationship between social exclusion and Facebook addiction. The study yielded several important findings, which have the following implications. First, social exclusion in Facebook is positively associated with Facebook addiction. Second, when social exclusion occurs on Facebook, the higher the degree of narcissistic grandiosity is, the stronger the Facebook addiction is. Third, when social exclusion is a factor that promotes Facebook addiction, surveillance use plays a mediating role to monitor social information between social exclusion and addiction.

## Figures and Tables

**Figure 1 ijerph-16-03813-f001:**
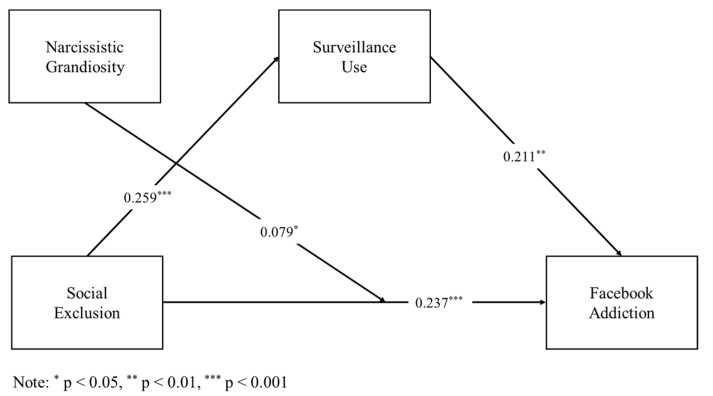
The indirect effect of surveillance use modeling and the conditional direct effect of narcissistic grandiosity on the relationship between social exclusion and Facebook addiction.

**Figure 2 ijerph-16-03813-f002:**
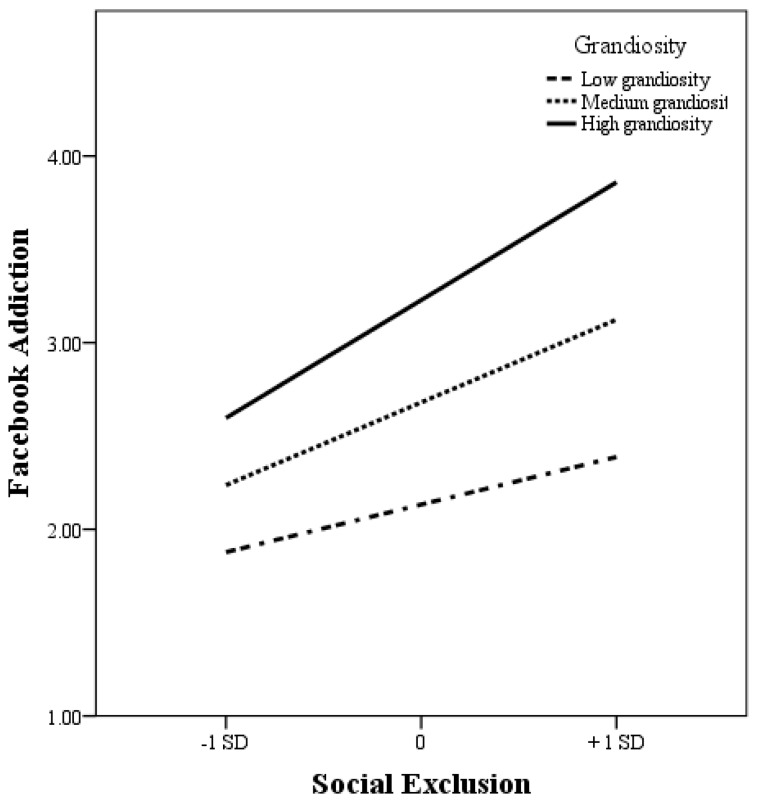
The conditional direct effect of narcissistic grandiosity on social exclusion and Facebook addiction.

**Table 1 ijerph-16-03813-t001:** Means, standard deviations, correlations, and alpha reliabilities ^a^ for variables.

Variables	M	SD	1	2	3	4
Social Exclusion	3.545	1.612	(0.921)			
Surveillance Use	4.026	1.273	0.328 **	(0.752)		
Facebook Addiction	2.756	1.434	0.515 **	0.455 **	(0.867)	
Grandiosity	3.131	1.333	0.405 **	0.421 **	0.610 **	(0.921)

Note: ^a^ on diagonal in parentheses, ** *p* < 0.01.

**Table 2 ijerph-16-03813-t002:** Conditional direct association of social exclusion on Facebook addiction (N = 188).

Narcissistic Grandiosity	Effect (Slope for Social Exclusion)	SE	*t*	*p*	LLCI	ULCI
Low (1.8)	0.131	0.064	2.073	0.040	0.006	0.257
Medium (3.13)	0.237	0.053	4.508	<0.001	0.133	0.341
High (4.46)	0.342	0.070	4.865	<0.001	0.203	0.481

Note: SE (Standard Error), LLCI & ULCI (Low and Upper Levels for Confidence Interval).

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
