# Peer review of "Social Exclusion, Surveillance Use, and Facebook Addiction: The Moderating Role of Narcissistic Grandiosity"

_ijerph, 2019, doi:10.3390/ijerph16203813_

Round 1
Reviewer 1 Report
Thanks for the possibility to revise the manuscript titled "Social exclusion, surveillance use, and Facebook addiction: The moderating role of narcissistic grandiosity". I read the manuscript with great interest and I think it adds to the existing literature. I, therefore, believe that the article should be accepted for publication in this journal, after minor revisions.
Specifically, while the introduction to the work does an excellent review of the literature, explaining and explaining the assumptions made, the section "Materials and Methods" should be enriched with more specific information.
It would be important to give more information on sample retrieval. Also, is there any additional demographic information available for the sample?
The author stated that "The average survey completion time was 3.51 minutes" (line 177). Does this mean that the participants took about 6 seconds for each item? Do you think that the time taken is adequate?
Furthermore, the author state that "after the completion of the survey, they were compensated appropriately for their participation" (lines 179-180). How?
On line 196, the author reported an alpha value of 0.921 and on line 200, the author reported an alpha value of .725. Be consistent in the way you present decimals.
On line 214, there's a typing error. Delete the point after the word "used". The author is also invited to discuss the various results that have emerged from the literature more widely and, above all, the implications that the results may have.
Reviewer 2 Report
The present paper addresses an interesting topic. However, I identified some limitations:
Literature review (Introduction and Hypotheses development). Please add and discuss specific studies that are consistent with the aim of the research.
2.4. The mediating role of surveillance use. Please justify the mediation effect from a theoretical point of view.
Please clarify the following sentence: “After the completion of the survey, they were compensated appropriately for their participation”.
3.2. Measures. Please describe the language of each measure and their validity.
I think could be better to report the standardized scores.
Reviewer 3 Report
The present study examined associations between social exclusion and Facebook addiction through surveillance use and moderation by narcissistic grandiosity. While the topic is of interest, I have the following queries and concerns for the author(s) to address:
1. As written the abstract is a little vague and unclear in parts. Greater details on the participant demographics should be included, and the model/statistical outcomes are not mentioned.
2. Introduction – ln 39- introduces the study aims before key details and past literature have been examined. This could be deleted as the relevant details are provided later in the introduction.
3. Introduction ln 58-references for the assertions made in this paragraph related to social exclusion leading to Facebook addiction are needed.
4. Introduction – line 71 “Facebook-related stress” is introduced nut not adequately defined? Also how does this concept relate to fear of missing out?
5. Methods – line 183 – Can you provide a break-down of which relationships were chosen for the imaginary interaction e.g., romantic partner, co-worker, colleague etc. It would seem that this introduces great variability into the study that is difficult to control.
6. Related to point 5 above – was there any relationship between who the participant chose for the imaginary interaction for X and other variables such as age, gender, marital status etc.
7. Methods – which items of the perceived risk of exclusion scale were left off the current protocol, and how did this affect the scale’s reliability compared with its use in previous studies?
8. ‘Addiction’ is used inconsistently throughout the paper, while addictive tendencies are used in the appendix? Can the authors please clarify the relevant construct?
9. Results – should include a table of relevant descriptive data of the sample. This is eluded to in the first paragraph of the results but not included in sufficient detail.
10. Results – line 235 – can effect sizes for the relevant regressions be included? This should also be accompanied with a statement of the generalisability of study results to other samples given the relatively small sample size studied.
11. Table on page 7 ln 263 reports p values of .000, however this is not possible. This should be changed to <.001
12. Discussion ln 305 – different types of relationships are discussed here but in very limited detail. How are different relationships likely to affect outcomes?
13. Discussion – treats study outcomes as representing a one-to-one relationship, however facebook users typically have many friends (each which may have different privacy settings applied to them based on their relationship). How is this likely to affect results given that embedded within these facebook friendships will be some close relationships (e.g., romantic partner) and some acquaintances (e.g., work colleagues)?
14. While participants were asked to imagine interactions for the study purposes, were they also asked how often they had engaged in surveillance using social media in real-life?
15. Facebook is often used in conjunction with other social media sites – so a sentence or two talking about the generalisability of results to other SNS platforms should be included.
Round 2
Reviewer 3 Report
The authors have addressed all of the comments that I raised during the review process appropriately.